# Reinforcement Learning Environment for Wavefront Sensorless Adaptive Optics in Single-Mode Fiber Coupled Optical Satellite Communications Downlinks

Payam Parvizi [1], Runnan Zou [1], Colin Bellinger [2], Ross Cheriton [2] and Davide Spinello [1,*]

1  Department of Mechanical Engineering, University of Ottawa, Ottawa, ON K1N 6N5, Canada; pparv056@uottawa.ca (P.P.); rzou043@uottawa.ca (R.Z.)
2  National Research Council of Canada, Ottawa, ON K1A 0R6, Canada; colin.bellinger@nrc-cnrc.gc.ca (C.B.); ross.cheriton@nrc-cnrc.gc.ca (R.C.)
*  Correspondence: dspinell@uottawa.ca; Tel.: +1-613-562-5800 (ext. 2460)

**Abstract:** Optical satellite communications (OSC) downlinks can support much higher bandwidths than radio-frequency channels. However, atmospheric turbulence degrades the optical beam wavefront, leading to reduced data transfer rates. In this study, we propose using reinforcement learning (RL) as a lower-cost alternative to standard wavefront sensor-based solutions. We estimate that RL has the potential to reduce system latency, while lowering system costs by omitting the wavefront sensor and low-latency wavefront processing electronics. This is achieved by adopting a control policy learned through interactions with a cost-effective and ultra-fast readout of a low-dimensional photodetector array, rather than relying on a wavefront phase profiling camera. However, RL-based wavefront sensorless adaptive optics (AO) for OSC downlinks faces challenges relating to prediction latency, sample efficiency, and adaptability. To gain a deeper insight into these challenges, we have developed and shared the first OSC downlink RL environment and evaluated a diverse set of deep RL algorithms in the environment. Our results indicate that the Proximal Policy Optimization (PPO) algorithm outperforms the Soft Actor–Critic (SAC) and Deep Deterministic Policy Gradient (DDPG) algorithms. Moreover, PPO converges to within 86% of the maximum performance achievable by the predominant Shack–Hartmann wavefront sensor-based AO system. Our findings indicate the potential of RL in replacing wavefront sensor-based AO while reducing the cost of OSC downlinks.

**Keywords:** wavefront sensorless adaptive optics; reinforcement learning; optical satellite communications downlinks; fiber coupling



## 1. Introduction

Optical beam wavefronts can experience distortion as they propagate through atmospheric turbulence, as illustrated in Figure 1. This distortion can lead to a reduction in the potential bandwidth of the link [1,2]. Currently, most optical satellite communication ground stations direct optical beams onto photodetectors or couple them into multi-mode fiber to improve coupling efficiency [3]. However, this approach comes with limitations, as it restricts the use of highly efficient phase modulation schemes as well as the use of fiber amplifiers for long-haul data transmission. Coupling into single-mode fiber can enable >10 Gbps data rates per wavelength channel, but it requires much higher optical powers from the satellite to compensation for reduced coupling efficiencies. Larger telescope apertures provide little benefit since they are more affected by atmospheric turbulence, leading to diminishing returns on the signal. The issue of atmospheric turbulence has been successfully mitigated in astronomy using adaptive optics (AO) [4–7]. The traditional AO system dynamically corrects the wavefronts using a deformable mirror through closed loop feedback from a wavefront sensor.

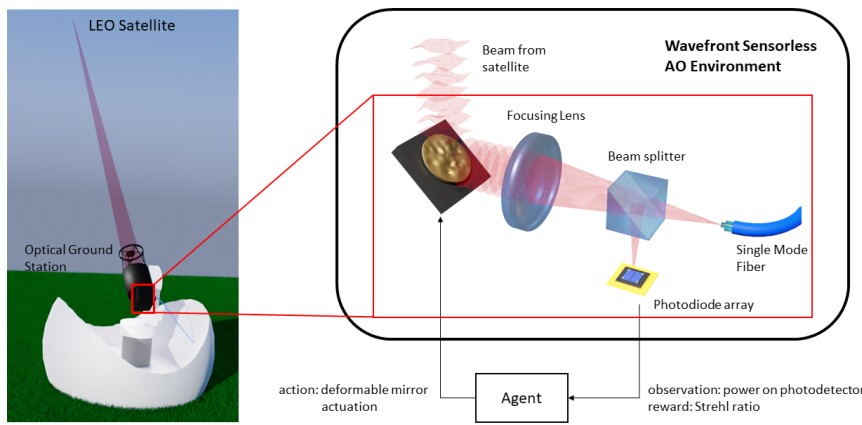

**Figure 1.** Schematic of the RL environment of wavefront sensorless AO system.

Traditional AO systems are still costly and complex, with a significant portion of the cost arising from the wavefront sensor, especially when dealing with infrared beams in optical satellite-to-ground links which exhibit higher read noise and require cooling unlike silicon cameras [8]. Furthermore, they consume a significant fraction of the incident beam intensity to adequately illuminate the camera pixels and add latency to the system. This latency can lead to outdated wavefront measurements in fast moving low earth orbit (LEO) satellites, resulting in errors at the space and time scales of optical coherence [9]. Recently, research has demonstrated the potential of reinforcement learning (RL) to tackle complex control problems in various domains of AO [10], such as astronomy [11–13], wavefront sensor-based systems [14–17], wavefront sensorless AO systems [18–20], and microscope systems that, unlike atmospheric turbulence, deal with lower-speed turbulence caused by aqueous solutions and optical aberrations in a microscope applications [21]. The existing implementations of RL on AO systems are not optimized for optical satellite communication due to their primary focus on optimizing image sharpness rather than improving the reliability of optical data links. In other words, although image sharpness can be used for fiber-coupling efficiency, it gathers excessive information that can compromise loop speed. Ensuring a stable and dependable optical link is prioritized in moving towards a consistent and uninterrupted high-data transmission with lower latency and cost. A low pixel count detector mitigates high-latency and expensive read-out circuits, such as those found in infrared cameras. Additionally, image-sharpness-based implementations are specifically customized for optimizing wavefront distortion under the long coherence time conditions found in environments like microscopy and ophthalmology.

In RL, agents' decisions are based on the information observed from the environment. However, the observations received by an agent might not contain all the necessary information for decision-making, leading to a partially observable environment. Control methods for partially observable dynamic environments are an outstanding challenge [22]. Examples of applications that fall under these conditions are free-space communication [23], laser machining, astronomy [24], retinal imaging, microscopy, real-time video upscaling, underwater imaging and communication, and even real-time music mixing/production, where lower observability can lead to reduced latencies and cost in time-sensitive applications. This research aims to realize fast, reliable, and lower-cost satellite communications downlinks, one of the most challenging of these applications.

It is estimated that for every 1% increase in light coupling, the system cost is decreased by 2% compared to the original cost [25–27]. In a traditional AO system, opting for a 40 cm telescope instead of a 60 cm one leads to substantial savings. This includes a minimum of USD 50,000 in the mount and telescope system [27], a minimum of USD 30,000 in the wavefront sensor, and a minimum of USD 20,000 in the dome enclosure. In contrast to traditional AO, incorporating an RL model into the system would only require a compact processor such as a field-programmable gate array (FPGA) and a simpler detection unit

like a quadrant photodetector, which would be significantly more cost-effective and offer minimal latency.

Given the use of a low-dimensional quadrant photodetector, we have partial observability of the environment [28–31], which can pose a challenge for RL algorithms to provide optimal policies, particularly when dealing with the high-dimensional action spaces of the deformable mirrors [32]. Furthermore, atmospheric turbulence is highly unpredictable and can vary from low to high turbulence conditions. It is crucial for RL algorithms to be adaptable and sample-efficient under a diverse range of turbulence conditions to ensure stable and accurate results.

To address these challenges, we conducted an analysis of RL algorithms within two distinct environmental scenarios: quasi-static and semi-dynamic environments. In the quasi-static environment, it is assumed that the turbulence profile is static during training, corresponding to the condition that the RL model convergence is significantly faster than the atmospheric coherence time. Such a condition can be relevant in emerging photonic adaptive optics concepts [33] and in cases where the ground layer turbulence coherence time is long. On the other hand, in the semi-dynamic environment the turbulence profile remains static in each episode and randomly changes in the next episode. The choice of a semi-dynamic environment is motivated by its utility in generating initial neural network weight configurations before transitioning to a dynamic environment, which is a closed representation of operational conditions. For this reason, we analyzed RL algorithms in quasi-static atmospheric turbulence conditions and we compared the RL environment configurations used in the quasi-static atmosphere with a new configuration in the semi-dynamic environment, defined by changes in observation space, action space, and reward function. This comparison was executed using an on-policy RL algorithm and the widely used Shack–Hartmann wavefront sensor-based AO that measures the displacement of the focal spots as a distorted wavefront propagates through the lenslet array [34].

This paper reports on the first phase of a three-phase project that includes *(a)* developing RL algorithms for wavefront sensorless AO in a simulated atmosphere, *(b)* characterizing the RL algorithms through simulations, and *(c)* deploying the RL model in a real-world AO system. In this proposed setup, the RL agent learns to directly control the deformable mirror using the power distribution on the focal plane. Figure 1 illustrates the proposed RL environment.

In our empirical analysis, we compare Soft Actor–Critic (SAC) [35], Deep Deterministic Policy Gradient (DDPG) [36], and Proximal Policy Optimization (PPO) [37] deep RL algorithms to an idealized traditional AO system with a Shack–Hartmann wavefront sensor. For further details on the deep RL algorithms mentioned, please refer to Section 4.1, Section 4.2 and Section 4.3, respectively. Our results suggest that RL can enable moderately efficient coupling into single-mode fiber without a wavefront sensor in quasi-static and semi-dynamic atmospheric conditions.

To summarize, the contributions of this work are:

- The development of a simulated wavefront sensorless AO-RL environment for training and testing RL algorithms. This is the first AO-RL environment that is implemented according to the standards of the OpenAI Gymnasium framework which simplifies the analysis of RL algorithms. The related source code link can be found in the 'Data Availability Statement' section.
- The first demonstration of the potential for RL in wavefront sensorless AO satellite data downlinks.

The remainder of the paper is structured as follows. Section 2 includes background information on AO and related work on RL in the context of AO, while Section 3 details the RL environment developed as part of this work. The experimental setup and RL algorithms are described in Section 4, and Section 5 presents the results. In Section 6, we discuss the limitations of the RL environment and the results. Section 7 includes summarizing remarks and current and future work.

## 2. Background

### 2.1. Adaptive Optics (AO)

The objective of AO is to eliminate any spatial phase distortions in an optical wavefront. The AO method was first proposed by Babcock [38] to improve observational astronomical by correcting wavefronts with a deformable optical element controlled by a wavefront sensor. New techniques and results have since been consistently published, primarily focusing on advancements in wavefront sensors, deformable mirrors, and control algorithms [39–43].

After propagating through the atmosphere to the telescope, light is distorted by subtle changes in the temperature and pressure (and hence the index of refraction) of the air, which varies in time and space [44]. The corresponding wavefront profile is measured by a wavefront sensor, which informs a control command for a deformable mirror to flatten the wavefront.

### 2.2. Satellite-to-Ground Optical Communication

To enable satellite-to-ground optical communication, low earth orbit satellites are being equipped with laser transmitters. Line of sight is required for transmission between the telescope and a satellite and is only maintained for a few minutes during the satellite pass, after which the telescope must reposition to track another satellite.

The atmosphere has a characteristic turbulence timescale on the order of $\sim$1 ms, which varies significantly with the satellite's elevation angle and turbulence conditions. This value can be as short as 0.2 ms and as high as 10 ms. Turbulence is generally the strongest at the lowest elevations due to the effective thickness of the atmosphere. However, the ground layer has a longer coherence time when compared to the high-altitude jet stream where the timescales are much shorter. The optical path from the satellite to the telescope is also changing due to the continuously changing satellite elevation angle. If an approximate solution can be found within a few milliseconds, the atmosphere can be considered to be in a quasi-static state and a static turbulence profile for the purposes of training can be considered valid. This paper is focused on optical ground stations near sea-level and therefore we adopt 1 ms as the target coherence timescale, neglecting the rapidly changing high altitude optical path column and high wind speeds.

## 3. Wavefront Sensorless AO-RL Environment

The RL environment is implemented according to the standards of the Open AI Gymnasium framework [45]. The HCIPy: High Contrast Imaging for Python package [46] serves as the foundation of the RL environment. HCIPy offers a comprehensive set of libraries related to AO, including wavefront generation, atmospheric turbulence modeling, propagation simulation, fiber coupling, implementation of deformable mirrors, and wavefront sensors. A simulated AO-RL environment is a critical first step in the process of developing RL-based wavefront sensorless satellite communications downlink systems. It enables one to assess and refine the RL to meet the strict requirements of this domain prior to costly evaluation in physical simulations and the real world.

The AO system simulated in this environment couples 1550 nm light into a single-mode fiber under various turbulence conditions which are characterized by the parameter $D/r_0$ (as shown in Figure 2). This parameter represents the ratio of the telescope's diameter ($D$) to the Fried parameter ($r_0$) and it serves as a measure of the quality of optical transmission through the atmosphere. A lower Fried parameter leads to more pronounced wavefront distortions. In this work, we maintain a constant $D$ value of 0.5 m, while the $r_0$ value is adjusted to evaluate the system's performance under varying atmospheric turbulence levels. It is assumed that the atmospheric turbulence is either quasi-static or semi-dynamic, and the satellite remains in a fixed position throughout the experiments.

**Figure 2.** Atmospheric turbulence conditions with respect to $D/r_0$.

A graphical presentation of the RL environment is displayed in Figure 1. The simulation environment updates in discrete time steps for practical purposes and to maintain consistency with the standard RL framework. At each time step $t$, the RL agent receives an observation of the system's current observation ($o$) and a reward ($r$). The observation encodes the power after the focal plane in the AO system, while the reward is computed by utilizing the power distribution after the focal plane. Based on the agent's parameterized policy, $\pi_\theta$, and the current observation, the agent selects the next action $\pi_\theta : o \rightarrow a$. The agent's actions control the deformable mirror in the AO system. When controlled optimally, the incoming optical beam becomes concentrated and centered on the single-mode fiber.

*3.1. Episodic Environment*

In real-world scenarios, this RL problem can be characterized as either a finite or infinite horizon problem. In the finite horizon case, each episode lasts for the duration of the satellite's communication with the receiver. Here, we focus on the episodic form of the problem. We assess the effectiveness of an RL policy that can map the deformable mirror from its neutral position (flat mirror) to a shape that focuses the beam on the single-mode fiber. We conducted a parametric study with respect to the episode's length, ranging from 10 to 100 time steps. Our findings indicate that the adoption of an episode length of 20–30 time steps is sufficient to achieve optimal action.

In the generated AO-RL environment, the option of selecting quasi-static, semi-dynamic, or dynamic environments is made available. As previously mentioned in Section 2.2, in a quasi-static environment, it is assumed that the atmosphere can be considered to be in a quasi-static state and a static turbulence profile for training purposes is considered valid. In a semi-dynamic environment, the configuration of the quasi-static atmosphere changes in each episode. The utilization of a semi-dynamic environment can be found to be helpful in the generation of the initial configuration of neural network weights before transitioning to a dynamic environment. In a dynamic environment, the movement of the atmosphere is influenced by a velocity vector where the final timestep of episode $i$ is the same as the initial timestep of episode $i + 1$.

*3.2. Action Space*

In the context of AO, the actions of the RL agent involve movements of the actuators located beneath the deformable mirror, as shown in Figure 3. The number of actuators determines the degree of freedom of the mirror's shape. The actuators are responsible for controlling the continuous reflective surface of the deformable mirror. The range of movement for each actuator is $\pm 5$ μm, providing a high degree of precision in the mirror's surface shape and position.

According to Tyson [6], the wavefront is a 2D map of the phase at a plane that is normal to the line of sight from the origin of the beam to the target. In the field of AO, multiple methods are utilized to represent this 2D wavefront map, including the power-series representation [6] and the Zernike series [47–49]. An effective approach to analyzing phase aberrations involves decomposing the wavefront into a series of polynomials, which form

a complete set of orthogonal polynomials defined over a circular pupil [47,50]. This series comprises sums of power series terms with appropriate normalizing factors [49,51].

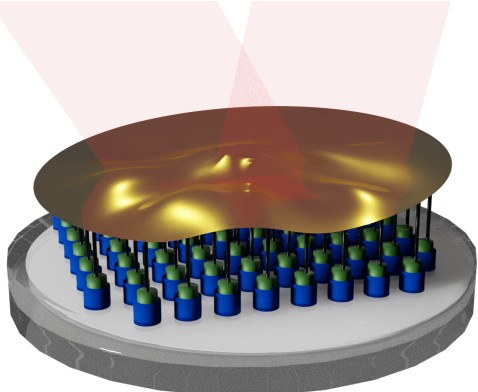

**Figure 3.** Illustration of a deformable mirror surface and incident beam.

Our RL environment presents two choices within the action space: the number of actuators or Zernike polynomials [47]. Given our available setup, the action space can be configured either directly with 64 actuators or with deformable mirror Zernike polynomials truncated at the first six modes (second radial orders) in a circular geometry. The decision to use six-mode Zernike polynomials stemmed from experiments conducted with Shack–Hartmann wavefront sensor-based AO. These experiments demonstrated that employing six modes of Zernike polynomials is adequate to achieve a Strehl ratio of more than 50% under average atmospheric turbulence conditions (where $D/r_0 < 5$).

In the former case, a 64-actuator segmented deformable mirror is simulated in a quasi-static environment, with eight actuators along each linear dimension. As a result, the RL agent is tasked with selecting actions from a 64-dimensional action space, where each actuator can be independently and continuously adjusted within a certain motion range. This setup allows for precise and smooth control of the deformable mirror. The actuation has a limit corresponding to the maximum optical phase error that is possible under the atmospheric conditions used in training. High-dimensional action space can potentially pose difficulties for RL algorithms in non-static environments due to the curse of dimensionality. For this reason, within our RL environment, we implemented the dimensionally reduced action space based on the Zernike series [47], which can be obtained from the observation space representing the wavefront.

It is assumed that the deformable mirror operates at a sufficiently fast speed to allow for approximating the atmosphere as quasi-static, given that most deformable mirrors are capable of correction at speeds of up to 1–2 kHz [52]. Faster deformable mirrors can be achieved using smaller mirrors. For a 50 cm telescope and this deformable mirror choice, the system can be expected to achieve reasonable correction for turbulence conditions of less than $r_0 = 6.25$. We expect $r_0$ conditions to range from 5 cm to 15 cm for satellite elevation angles above 15°.

### 3.3. Observation Space

We utilize the power of the wavefront that propagates through the focal plane to form the observation of the state of the environment. The focal plane is shown on the left in Figure 4. The white circle within the figure indicates the entrance of the single-mode fiber.

The observable states of the system directly and efficiently relate to the light coupled into the fiber through a reward function calculation, as explained in Section 3.4. We choose to rely on this, rather than using the full Markovian states, which would require access to information about the satellite's angle and atmospheric conditions. In other words, observations rely on the information directly measured about the state of the environment after the wavefront has been propagated through the focal plane and discretized into a sub-

aperture array of $n \times n$ pixels, whereas a Markovian state requires complete information about the system to fully describe it, which would introduce high levels of uncertainty since some of the quantities are not directly measurable or measured. The trade-off is the loss of full observability due to the compression of the state for cost and latency. Nonetheless, RL has tools to handle partial observability [53] and our agents can learn an effective policy utilizing the photodetector represented in Figure 4 (right), with each red square representing a pixel of the photodetector.

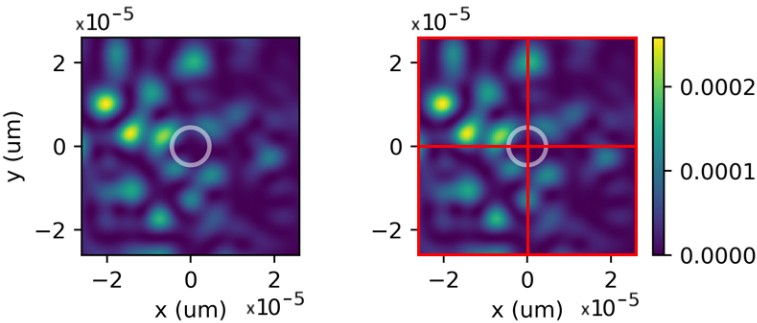

**Figure 4.** Focal plane profile, (**left**) continuous, (**right**) discretized.

In our RL environment, we can choose the dimension of the observation space, which is $n^2$. One might increase the value of $n$ to increase the dimension of the observation space in order to decrease the partial observability.

We discretize the focal plane into a sub-aperture array consisting of $2 \times 2$ pixels under quasi-static atmospheric conditions. This setup can be implemented using a fast and relatively cost-effective quadrant photodetector, as illustrated on the right in Figure 4. The use of a low-pixel count detector mitigates the need for slower and more expensive read-out circuits used in infrared cameras, allowing for more light per pixel and reducing noise, which, in turn, improves the speed of the RL algorithm training. However, a $2 \times 2$ sub-aperture array may not suffice for non-static atmospheric conditions due to low-partial observability and, in such cases, the dimension of the observation space can be increased by increasing the value of $n$.

*3.4. Reward Function*

There are two available choices for the reward function in the RL environment: the Strehl ratio and the combination of single-mode fiber total power with the Structural Similarity Index (SSIM).

In the former, the reward function is calculated using the Strehl ratio of the optical system, defined as the ratio of the normalized peak intensity of the point spread function to the peak intensity of the ideal point spread function without aberrations. A high Strehl ratio implies a high degree of wavefront correction, where a focused beam of light resembles an Airy disk and is approximately proportional to the amount of light that can be coupled into a fiber [54]. It is considered an approximation because, for optimal coupling into an optical fiber, a focused beam should resemble a Gaussian profile. As proposed by Mahajan [55], the Strehl ratio $r_1$ of systems with a circular pupil is expressed in terms of the variance of the phase aberration across the pupil

$$r_1 = e^{-\sigma_\Phi^2} \tag{1}$$

where $\sigma_\Phi^2$ represents the variance of the phase aberration.

Although the Strehl ratio is commonly used in AO, its computation is not ideal for the applications in this work since it requires a point spread function and a camera, whereas our system explicitly avoids reliance on a camera (specifically an InGaAs camera), in favor of a cheaper, compact, and faster solution. The calculation of the Strehl ratio requires sufficiently long exposures for speckle variation and there are alternative methods that utilize fewer

array sizes. Additionally, the Strehl ratio is not an ideal metric for coupling into a single-mode fiber, as the correlation between the single-mode fiber-coupling efficiency and the Strehl ratio is reduced at high Strehl ratios, restricting its applicability as reward function to lower values. At higher Strehl ratios, it is limited by the mode mismatch between a focused flat top beam and the Gaussian mode in a single-mode fiber [54,56]. Therefore, a different reward function has been added to the RL environment. This additional reward combines the total power of single-mode fiber and the Structural Similarity Index (SSIM). Although the total power of a single-mode fiber is a crucial indicator of its ability to concentrate power, due to its very small diameter, it is not particularly useful for exploration as it misses a significant portion of observations. Therefore, relying solely on the total power of a single-mode fiber as a reward function is inadequate. To address this, we combine it with SSIM as a criterion to measure the similarity between two arrays on a photodetector, like the Strehl ratio. The first array serves as a reference, which represents the power distribution when there is no wavefront distortion, and the second array represents the current power distribution. In other words, single-mode fiber total power measures the performance of the exit of the single-mode fiber, while SSIM measures the performance of the entrance of the single-mode fiber. This reward, named $r_2$, is formally defined by

$$r_2 = \beta P_{SMF} + (1 - \beta) \frac{(2\mu_{curr}\mu_{ref} + c_1)(2\sigma_{curr,ref} + c_2)}{(\mu_{curr}^2 + \mu_{ref}^2 + c_1)(\sigma_{curr}^2 + \sigma_{ref}^2 + c_2)} \tag{2}$$

where $\beta$ is a constant that controls the weighting between single-mode fiber total power ($P_{SMF}$) and SSIM. The SSIM equation involves the variables ($\mu_{ref}, \sigma_{ref}$), which are the mean and variance of the power distribution of the reference array when there is no distortion, and ($\mu_{curr}, \sigma_{curr}$) as the mean and variance of the current power distribution. $c_1$ and $c_2$ are two constants to stabilize the division when the denominator is small, avoiding numerical singularities.

## 4. Methodology and Algorithms Training

In Section 3.1, we discussed our three distinct environment options: quasi-static, semi-dynamic, and dynamic environments. For the quasi-static environment, we used a configuration with a $2 \times 2$ observation space and a 64-actuator action space, along with a Strehl ratio reward function (1). In preparation for the dynamic environment analysis, we utilize a semi-dynamic environment with a configuration of $5 \times 5$ observation space and a first six modes (second radial orders) of Zernike polynomials action space, with a new reward function (2). In this section, we present the hyperparameter optimization of the RL algorithms in the quasi-static environment, using the former configuration.

In a quasi-static environment, we quantify the performance of RL algorithms by calculating the mean and the standard deviation of the reward function across 20 independent trials. Each RL algorithm has its hyperparameters tuned to the environment and the best-performing setup is compared to an idealized AO system with a Shack–Hartmann wavefront sensor.

We have implemented and compared three families of RL algorithms: Soft Actor–Critic (SAC), Deep Deterministic Policy Gradient (DDPG), and Proximal Policy Optimization (PPO). These three families cover on-policy and off-policy learning, stochastic and deterministic policies, and the use of entropy-based methods. These variations are essential when considering their applicability in the context of wavefront sensorless AO, as each approach has its strengths and weaknesses. In particular, the off-policy algorithms, SAC and DDPG, generally have better sample efficiency compared to on-policy algorithms. Alternatively, the on-policy algorithm, PPO, is often known for its stability and ease of training. Given sufficient time, on-policy algorithms can provide high-performance results. Moreover, the entropy regularization in SAC enables rich exploration, which can be beneficial when dealing with high-dimensional action spaces. Each algorithm is discussed in more detail in the following subsections. We present a comprehensive list of the

hyperparameters selected for each algorithm after tuning for a quasi-static environment in Table 1.

**Table 1.** Hyperparameters and corresponding values in quasi-static environment.

| Hyperparameter | SAC | DDPG | PPO |
|---|---|---|---|
| Buffer size | 128 | 256 | - |
| Actor—$lr$ | $5 \times 10^{-4}$ | $5 \times 10^{-5}$ | $1 \times 10^{-2}$ |
| Critic—$lr$ | $1 \times 10^{-2}$ | $1 \times 10^{-2}$ | $5 \times 10^{-6}$ |
| Actor—hidden dim. | 150 | 250 | 150 |
| Critic—hidden dim. | 80 | 65 | 50 |
| Clipping $\epsilon$ | - | - | 0.35 |
| Temp. $\alpha_t$-$lr$ | $1 \times 10^{-1}$ | - | - |
| Temp. $\alpha_t$-min limit | 0.4 | - | - |
| No episodes per iteration | 1 | 2 | 2 |
| No updates per iteration | 20 | 20 | 20 |
| Polyak ($\rho$) | 0.99 | 0.99 | - |
| Discount ($\gamma$) | 0.95 | 0.95 | 0.95 |
| Reward scaling | No | Mean–std | No |
| learned $\alpha_t$ | Semi | - | - |

*4.1. Soft Actor–Critic (SAC)*

SAC is an off-policy actor–critic algorithm based on a maximum entropy RL framework. It is particularly useful in complex and stochastic environments, such as wavefront sensorless AO systems. SAC uses a deep neural network to approximate the actor and critics. The actor component of the algorithm is tasked with maximizing the expected return while promoting exploration through random actions rather than becoming trapped in suboptimal policies. On the other hand, the critic component is responsible for estimating the Q-function of a given state–action pair. The Q-function provides feedback for improving the policy by adjusting the actions that the agent takes in each state to maximize the expected sum of future rewards [35].

SAC has shown promising results in various domains. However, one major drawback is its sensitivity to the choice of temperature $\alpha_t$ and intuitively selected target entropy parameters. These parameters play a crucial role in the algorithm's performance and their selection can significantly affect the outcome. To address this, Reference [57] proposed a method of automatic gradient-based temperature tuning by matching the expected entropy $\log \pi_t^*(a_t|s_t; \alpha_t)$ to a target entropy value $\bar{\mathcal{H}}$ at time $t$:

$$\alpha_t^* = \arg \min_{\alpha_t} \mathbb{E}_{a_t \sim \pi_t^*} \left[ -\alpha_t \log \pi_t^*(a_t|s_t; \alpha_t) - \alpha_t \bar{\mathcal{H}} \right], \qquad (3)$$

where the temperature $\alpha_t$ controls the stochasticity of the optimal policy, and $a_t$ and $s_t$ are the action and the state at time $t$, respectively.

In our preliminary assessment, we evaluate the SAC using three different temperature settings: fixed, learned, and semi-learned temperatures. Under the fixed temperature condition, we achieved the best reward by setting the constant value $\alpha_t = 0.4$. This resulted in a mean reward of 52.63% with a standard deviation of 14.64% at the end of the training process. For the learned temperature condition, we optimized it using a learning rate $\alpha_t$-$lr = 10^{-1}$ within the learning rates ranging from $10^{-6}$ to $5 \times 10^{-1}$. The semi-learned temperature condition was also optimized using the same learning rate, along with a minimum value of $\alpha_t = 0.4$, resulting in improved performance compared to the entire range of learning rates and minimum $\alpha_t$ values between 0 and 1. Figure 5 illustrates our results, showing that the fixed and semi-learned temperature settings show faster learning rates compared to the purely learned setting, and that semi-learned converges to the best policy. As a result, we have chosen to use the semi-learned temperature configuration in all subsequent experiments.

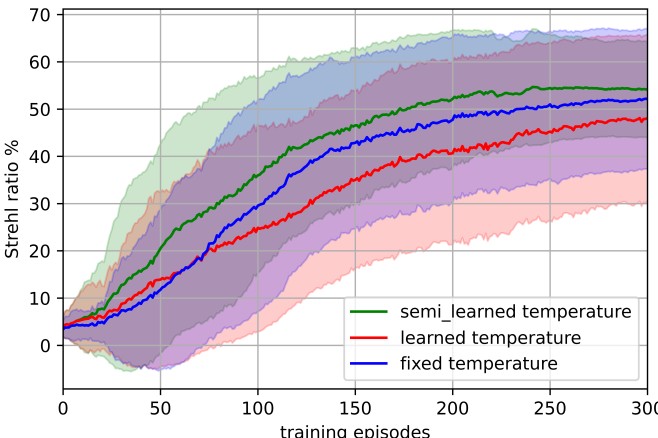

**Figure 5.** Comparison of the selection of the temperature ($\alpha_t$) in SAC applied on 20 randomly selected quasi-static atmospheric turbulences of $D/r_0 = 5$. Note that the shaded regions extend to negative values of the Strehl ratio because the standard deviation may be larger than the mean, represented by the solid curves.

### 4.2. Deep Deterministic Policy Gradient (DDPG)

DDPG utilizes a deep neural network to approximate the value function and policy, allowing it to handle high-dimensional observation spaces. This approach, as previously demonstrated by [36], can be effective in tackling complex environments. While DDPG has the advantage of ease of implementation, it can be sensitive to the choice of hyperparameters and can be prone to instability due to the choice of the reward function [58].

In our preliminary assessment, we compared the effect of reward normalization with the mean–standard deviation (mean–std norm) and the min–max norm method versus no normalization on the learned policy. The normalization process scales the rewards across episodes. This has been shown to help the model identify actions that lead to higher rewards, thus accelerating the algorithm's convergence. In addition, as the model reaches convergence, the variance of the rewards tends to decrease, making it more challenging for the model to adjust itself. By normalizing the rewards, the model can more effectively recognize these rewards and continue to make adjustments.

Figure 6 demonstrates that omitting reward scaling in this domain leads to slow convergence to a lower reward. Min–max norm and mean–std norm learn at similar rates; however, mean–std norm converges to a higher reward. Thus, mean–std normalization is employed for the subsequent experiments in our DDPG approach.

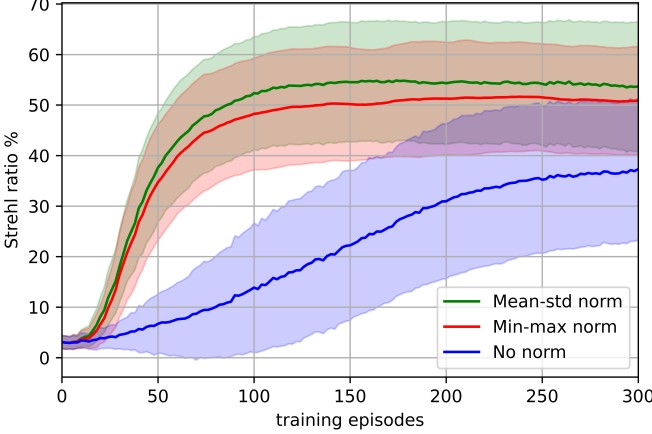

**Figure 6.** Comparison of the selection of the normalization technique in DDPG applied on 20 randomly selected quasi-static atmospheric turbulences with $D/r_0 = 5$.

### 4.3. Proximal Policy Optimization (PPO)

PPO is an on-policy, policy gradient algorithm. It alternates between sampling data through interactions with the environment and optimizing a surrogate objective function via stochastic gradient ascent [37]. PPO utilizes a deep neural network to approximate the policy and value function. PPO employs a "clipped surrogate objective" to penalize significant policy updates. The policy update, expressed as the probability ratio $r_t(\phi)$ between the old policy and the current policy at time $t$, is encoded in objective function

$$L^{CLIP}(\phi) = \hat{\mathbb{E}}_t[\min(r_t(\phi)\hat{\mathbb{A}}_t, clip(r_t(\phi), 1-\epsilon, 1+\epsilon)\hat{\mathbb{A}}_t)] \tag{4}$$

where $\phi$ is the vector of policy parameters. This technique limits eventual drastic changes in the policy during updates, and its effectiveness depends on the hyperparameter $\epsilon$, which controls the size of policy updates to prevent model collapse. Setting $\epsilon$ too small may result in slow convergence while setting it too large increases the risk of model collapse. The objective function limits the default policy gradient $r_t(\phi)\hat{\mathbb{A}}_t$ to the range of $[1-\epsilon, 1+\epsilon]\hat{\mathbb{A}}_t$, where $\hat{\mathbb{A}}_t$ is the advantage estimator.

Our preliminary analysis, as shown in Figure 7, demonstrates that PPO is robust to $\epsilon \in [0.05, 0.4]$ in this environment. Settings within this range show very subtle differences in variance, convergence rate, and convergence level. Generally, smaller $\epsilon$ values result in slightly slower convergence, whereas larger values converge faster but to marginally lower levels. Based on this analysis, all subsequent experiments have $\epsilon = 0.35$.

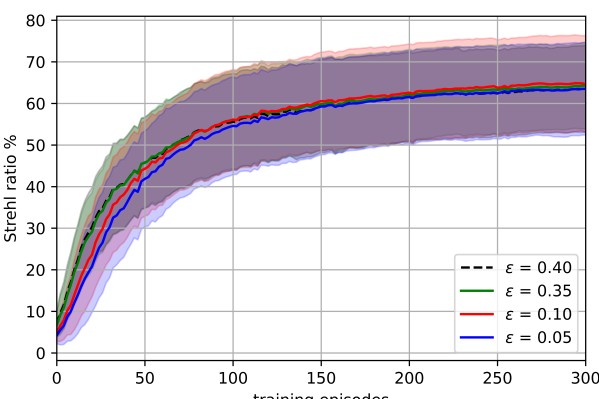

**Figure 7.** Comparison of the selection of clipping parameter $\epsilon$ in PPO applied on 20 randomly selected quasi-static atmospheric turbulence of $D/r_0 = 5$.

## 5. Results

The experiments were conducted using a computing system equipped with an Intel Core i7 processor and 16GB of RAM, running on a 64-bit Windows operating system. To facilitate the computation, several software libraries and frameworks were employed, including the Gymnasium environment version 0.29.1, the HCIPy framework version 0.5.1, and the Pytorch framework version 2.1.0, all of which were implemented in Python 3.9.6.

### 5.1. Comparison of RL Algorithms in Quasi-Static Environment

We used the light coupling performance obtained from Shack–Hartmann wavefront sensor data as the reference for comparison with the refined RL algorithms outlined in Section 4. This comparison was conducted under the quasi-static turbulence condition of $D/r_0 = 5$. The results are presented in Figure 8. Employing the Shack–Hartmann wavefront sensor with 12 lenslets across the aperture diameter for a total of 112 lenslets as a benchmark enabled us to comprehensively evaluate the effectiveness of the proposed RL algorithms in improving light coupling performance in the presence of atmospheric turbulence.

The SAC and DDPG algorithms rely on the randomness introduced by their replay buffers. To ensure that the results of these algorithms can be replicated, a fixed seed is used

as the initial parameter. To avoid the possibility of reusing the same set of random values in every iteration, the fixed seed value is incremented by 1 after each iteration.

The results shown in Figure 8 indicate that the PPO algorithm outperforms the SAC and DDPG algorithms in this experiment. Specifically, when we consider the randomly selected quasi-static atmospheric turbulence with a ratio of $D/r_0 = 5$, the PPO algorithm achieved a maximum reward of 67%, which is close to the maximum reward obtained by the Shack–Hartmann sensor, approximately 73%. Although SAC and DDPG still demonstrated acceptable performance with a maximum reward of 53% in the early training episodes, the PPO algorithm consistently demonstrated superior results. The improved performance of PPO over SAC and DDPG can be attributed to its ability to sample from the action distribution, which helps it avoid suboptimal local minima in high-dimensional observation spaces. In contrast, in SAC and DDPG, the actions that look nearly optimal can have an equal likelihood of being tried as those that appear highly suboptimal.

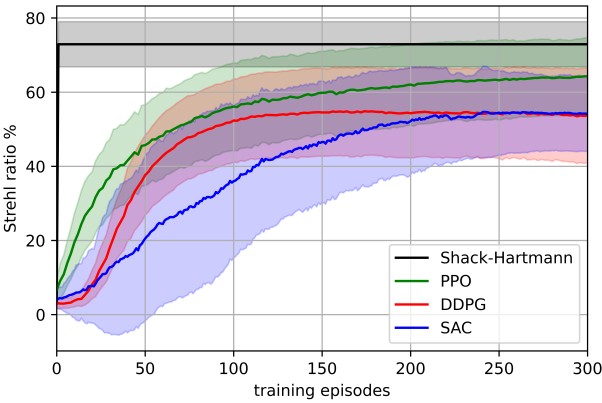

**Figure 8.** Comparison of algorithms applied on 20 randomly selected quasi-static atmospheric turbulences of $D/r_0 = 5$. Note that the shaded regions extend to negative values of the Strehl ratio because the standard deviation may be larger than the mean, represented by the solid curves.

5.1.1. Performance with Varying Turbulence Severity

In this section, we employ the PPO algorithm to evaluate its performance under low and high quasi-static turbulent conditions. The corresponding results are displayed in Figure 9.

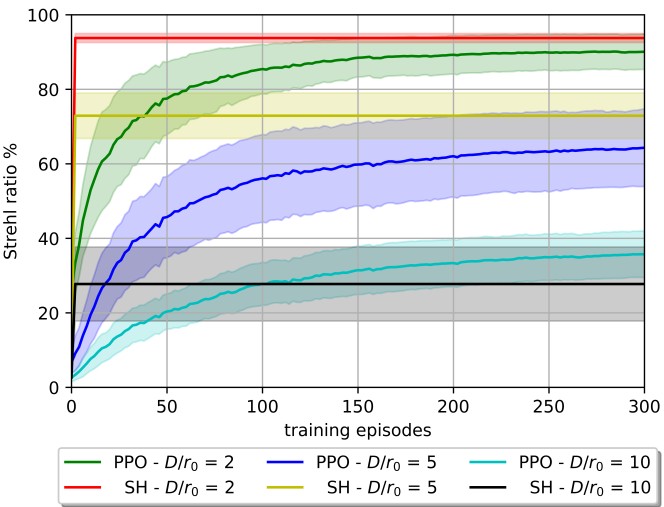

**Figure 9.** Average reward of 20 randomly selected quasi-static atmospheres of different $D/r_0$ ratios with 64 actuators and 4 observers on PPO algorithm and Shack–Hartmann wavefront sensor.

As anticipated and illustrated in Figure 9, a decrease in the Fried parameter value results in a decline in the model's capacity to achieve a higher reward. If the agent's performance cannot significantly improve beyond the uncorrected 2 to 10% Strehl ratio (depending on $D/r_0$ value), it can be considered impractical.

### 5.1.2. Power Distribution Comparison

The impact of using a PPO algorithm within an RL environment and using a Shack–Hartmann wavefront sensor on the power distribution of a wavefront under quasi-static turbulence condition of $D/r_0 = 5$ have been analyzed and illustrated in Figure 10. On the left side of Figure 10, the power distribution at the focal plane at the beginning of each episode when the wavefront is reflected through a flat deformable mirror. On the right side of the figure, we plot the results after: (upper right) the implementation of random actions through the PPO algorithm in the initial episodes, (middle right) the utilization of the Shack–Hartmann wavefront sensor, and (lower right) the application of the PPO algorithm following a sequence of episodes.

The utilization of the Shack–Hartmann wavefront sensor, as illustrated in Figure 10 (middle right), has resulted in a significant concentration of power at the center of the focal plane, with a Strehl ratio of approximately 70%. Similarly, the application of the PPO algorithm, as illustrated in Figure 10 (lower right), has also produced a significant concentration of power at the center of the focal plane, however, with a slightly lower Strehl ratio of approximately 60%.

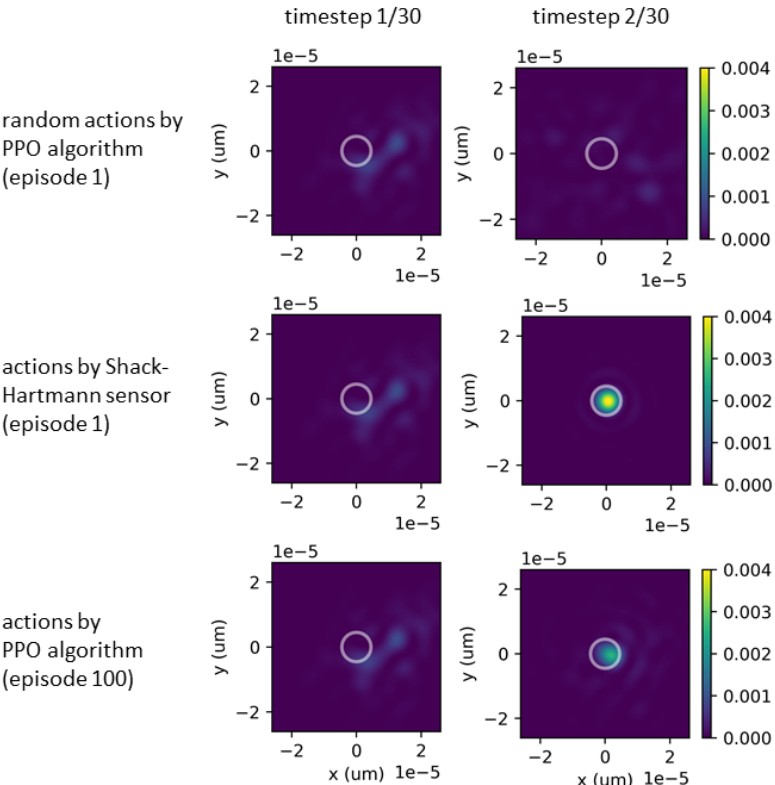

**Figure 10.** Power distribution on a focal plane (**left**) before proposed AO, (**upper right**) after the implementation of random actions through the PPO algorithm in the initial episodes, (**middle right**) after the utilization of the Shack–Hartmann wavefront sensor, (**lower right**) after the application of the PPO algorithm following a sequence of episodes.

### 5.2. PPO Algorithm in Semi-Dynamic Environment

The experimental setup assigned for a quasi-static environment consists of a 64-actuator action space, a $2 \times 2$ pixel observation space, and a Strehl ratio reward function. To extend

and validate our algorithmic framework for dynamic environments that better approximate real-world scenarios, such as the Earth's turbulent atmosphere, it is necessary to refine the RL environment to address specific challenges posed by non-static conditions.

Before transitioning to the dynamic environment, it is essential to adjust the initial weights of the agents to facilitate more effective and efficient learning. To accomplish this, we have employed a semi-dynamic environment, as explained in Section 3.1. This semi-dynamic environment employs the reward function $r_2$ in (2), replacing the Strehl ratio in (1) based on the discussion in Section 3.4.

In the conducted experiment, each iteration involves 100 episodes of a randomly selected quasi-static environment, with each episode consisting of 20 time steps. The results of our experiment on a semi-dynamic environment using a quasi-static configuration with the PPO algorithm are presented in Figure 11 (red). This shows that the configuration used for a quasi-static environment is insufficient for learning a policy within a semi-dynamic environment, reaching a mean value of 20% coupling efficiency. Based on this poor performance, we hypothesize that partial observability and the curse of dimensionality can be significant factors. Specifically, the observation space is limited to a low dimension of $2 \times 2$ pixels, which may result in inadequate information gathered from the environment, leading to partial observability. Additionally, the utilization of a high-dimensional action space can cause an exponential rise in computational effort and lead to the curse of dimensionality [60]. To address these challenges within our existing setup, we reduced the action space dimension by employing the first six modes (second radial orders) of Zernike polynomials and increased the observation-space dimension to a $5 \times 5$ pixel photodetector (config 2). Furthermore, considering the time dependency, we applied the frame stacking technique [59] using a sequence of three frames, obtaining the results in Figure 11 (blue). Hyperparameter optimization has been performed for this experiment.

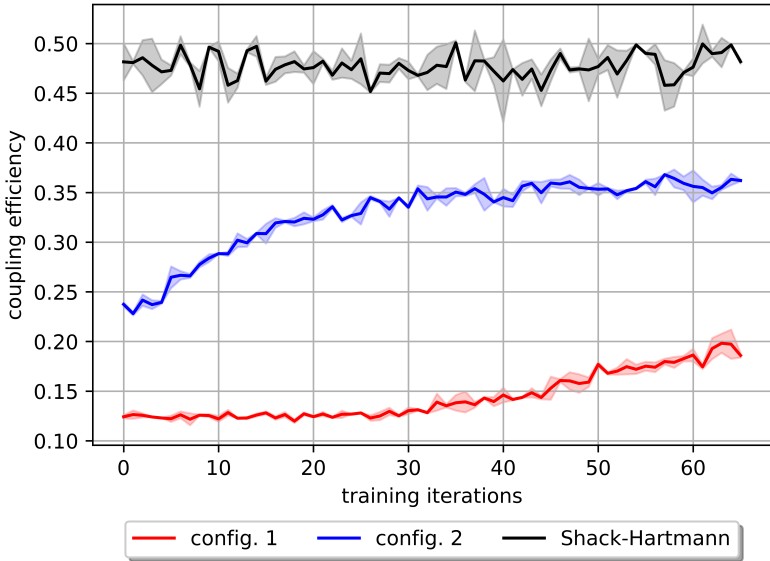

**Figure 11.** Comparison between two configurations in atmospheric turbulence of $D/r_0 = 3.33$ in semi-dynamic environment. Config. 1: observation space of $2 \times 2$ pixels, action space of 64 actuators. Config. 2: observation space of $5 \times 5$ pixels, action space of first 6 modes (2nd radial orders) of Zernike polynomials. The reward function is Equation (2).

By increasing the dimension of the observation space to a $5 \times 5$ pixel photodetector and reducing the action space to a first six modes (second radial orders) of Zernike polynomials, we can achieve a total power after single-mode fiber-coupling efficiency of 36%. In the semi-dynamic environment, it is noticeable that the Shack–Hartmann reaches a mean coupling efficiency of 48%. Part of the discrepancy could be attributed to the fact that the Shack–Hartmann wavefront sensor with 12 lenslets across the aperture diameter employs a

128 × 128 pixels camera for adjustment, in contrast to our use of a 5 × 5 pixel photodetector for calculating the reward value.

## 6. Discussion

In this study, we encountered a challenging scenario involving a high 64-dimensional continuous action space and a low-dimensional 2 × 2 observation space in quasi-static atmospheric turbulence. This situation could potentially hinder the effectiveness of RL algorithms. However, the successful performance of our RL algorithms in a quasi-static environment has demonstrated that their effectiveness does not solely depend on the limited dimensions of the photodetector and the deformable mirror. Instead, it arises from a combination of factors, including the robustness of the reward function, the quality of the employed photodetector, and the correlation within the action space. In other words, the Strehl ratio reward function used in a quasi-static environment offers a significant advantage due to its ability to provide high-quality rewards, which are calculated prior to the wavefront propagating through the photodetector, which had a higher pixel count (128 × 128 as opposed to 2 × 2) and resolution. Consequently, by employing a proficient function approximator and a robust reward function, the agent can effectively generalize over related actions. Given the large number of degrees of freedom in the mirror, there are numerous possible paths to focus the light and the agent only needs to cover one of these paths.

While increasing the dimension of the observation space and reducing the dimension of the action space was not necessary in quasi-static atmospheric turbulence, it became crucial in semi-dynamic atmospheric turbulence. For this reason, the observation-space dimension was changed to 5 × 5 and the action-space dimension was reduced to a first six modes (second radial orders) of Zernike polynomials. The reward function was changed from the Strehl ratio to Equation (2)) since the Strehl ratio cannot be accurately calculated using a 5 × 5 detection array, and to better correspond to coupling efficiency into a single-mode fiber. The new reward function is designed to perform calculations utilizing only a 5 × 5 pixel observation space.

The results of a quasi-static environment indicate that, while it is possible for the PPO, SAC, and DDPG RL models to determine an accurate set of deformable mirror actions, the results generally underperform the output of the Shack–Hartmann wavefront sensor. PPO outperforms the SAC and DDPG models, converging on a Strehl ratio of over 60% after hundreds of training episodes. One of the main limitations of these results is the applicability to a dynamic atmosphere with limited deformable mirror speeds. The 3 dB bandwidth of the fastest mechanical deformable mirrors is limited to <10 kHz, which implies that less than 10 action-measurement loop iterations are required for the quasi-static turbulence condition to hold. For most deformable mirrors, this requirement cannot be met. However, some faster deformable mirrors possess sufficient speed, and photonic chip-based phase corrector arrays are capable of speeds well in excess of 20 kHz [33] and can make use of a model requiring tens of action steps to converge.

Despite this limitation, we expect such simplifications to be reasonable as an RL-based system may still outperform a Shack–Hartmann wavefront sensor with its corresponding photon count requirements which are not considered in our comparisons. For existing and planned optical ground stations, AO may not be considered at all due to the cost, complexity, calibration, wavefront steps, aliasing, and latencies involved in Shack–Hartmann wavefront sensing. Therefore, any improvement to the wavefront beyond an uncorrected case is still of value. Furthermore, the application of models trained on quasi-static and semi-dynamic turbulence profiles may be of value to RL environments with dynamically changing turbulence profiles for improvement to signal re-acquisition times.

The choice to use 2 × 2 or 5 × 5 photodetectors as a source of feedback arises from their simplicity, cost-effectiveness, and high degree of correlation with improved single-mode fiber coupling. Our experiments using PPO showed rapid improvement in the reward, indicating that PPO can be an effective algorithm for wavefront correction under certain

conditions in quasi-static atmospheric turbulence. As demonstrated in Figure 9, although PPO performs better than the Shack–Hartmann wavefront sensor under high turbulence conditions, our results also revealed its performance may be limited due to the presence of a high-dimensional continuous action space and a low-dimensional continuous observation space. This can pose a significant challenge for the algorithm to explore the optimal control policy. These results emphasize the need for further research in developing the RL algorithms for partially observable environments to improve the wavefront correction performance in AO applications, particularly in high dynamic turbulence conditions.

## 7. Conclusions

We presented a cost-effective RL-based approach for wavefront sensorless AO in optical satellite communication downlinks. To gain a comprehensive understanding of the performance of the RL algorithms, we developed and shared the first optical satellite communications downlink RL environment. By using this RL environment, we implemented off-policy algorithms like SAC and DDPG, along with an on-policy algorithm, PPO, to optimize the coupling of 1550 nm light into a single-mode fiber under varying turbulence conditions.

In a quasi-static environment, our results showed that the PPO algorithm is particularly effective in achieving a high average correction with a low number of training iterations. Under atmospheric turbulence conditions with $D/r_0 = 2$ and 5, we observe that the PPO algorithm achieves performance close to 90% of Shack–Hartmann wavefront sensor-based AO with 12 lenslets across the aperture diameter. In more turbulent conditions, such as $D/r_0 = 10$, PPO outperforms the Shack–Hartmann wavefront sensor-based AO by approximately 10% Strehl ratio. This was accomplished while working with a low-dimensional observation space and a high-dimensional action space. This approach eliminates the need for wavefront sensor measurements, thereby reducing the cost and latency of optical satellite communications downlinks.

Towards the development of a real-world dynamic environment, we implemented a semi-dynamic environment, which can be helpful for initializing agents' weights for dynamic scenarios. The results of our experiment on a semi-dynamic environment using a quasi-static configuration with the PPO algorithm showed that the latter is insufficient for learning a policy within a semi-dynamic environment, reaching a mean value of 20% coupling efficiency. From this, we hypothesize that partial observability and the curse of dimensionality can have significant influence. To address these challenges while maintaining cost-effectiveness, we increased the observation-space dimension and reduced the action-space dimension by employing the Zernike series. This resulted in an average coupling efficiency of 36% compared to the Shack–Hartmann wavefront sensor-based AO, which reaches an average efficiency of 48%. Additionally, we observed that this configuration exhibits a faster response in the semi-dynamic environment compared to the previous configuration with lower dimensional observation space and higher dimensional action space.

Future work aims at developing robust RL models for wavefront sensorless AO in fully dynamic environments. Cost savings of 30% to 40% are expected through the use of smaller telescope systems and without the need for Shack–Hartmann wavefront sensors. This can be achieved by developing the framework of a deep RL model and training it to adapt to dynamically changing wavefront distortions across different satellite trajectories and deformable mirror actuator geometries. Our immediate focus involves investigating the performance of deep RL algorithms under various dynamic turbulence conditions within a simulated environment. Once this challenge is addressed, our next step involves training our model using the available setup, omitting wavefront sensors and cameras, but using low-pixel count photodetectors instead.

**Author Contributions:** Conceptualization, C.B., R.C. and D.S.; Methodology, P.P., R.Z., C.B. and R.C.; Software, P.P.; Validation, P.P., C.B. and R.C.; Formal analysis, P.P., C.B. and R.C.; Investigation, R.Z., P.P., C.B., R.C. and D.S.; Resources, C.B., R.C. and D.S.; Writing–original draft, P.P. and R.Z.; Writing– review & editing, P.P., C.B., R.C. and D.S.; Supervision, C.B., R.C. and D.S.; Project administration, C.B., R.C. and D.S.; Funding acquisition, C.B., R.C. and D.S. All authors have read and agreed to the published version of the manuscript.

**Funding:** This research was funded by the National Science and Engineering Research Council (NSERC) of Canada through Discovery grant RGPIN-2022-03921, and by the National Research Council (NRC) of Canada through the AI4D grant AI4D-135-2.

**Institutional Review Board Statement:** Not applicable.

**Informed Consent Statement:** Not applicable.

**Data Availability Statement:** The source code used in this work is available at the following GitHub Repository link: https://github.com/payamparvizi/adaptive_optics_gym (accessed on 5 December 2023).

**Conflicts of Interest:** The authors declare no conflict of interest.

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
