# Peer review of "Reinforcement Learning Environment for Wavefront Sensorless Adaptive Optics in Single-Mode Fiber Coupled Optical Satellite Communications Downlinks"

_photonics, doi:10.3390/photonics10121371_

Round 1

Reviewer 1 Report

Comments and Suggestions for Authors

The paper analyzes the Reinforcement Learning environment of wavefront sensorless adaptive optics in the downlink of single mode fiber coupled satellite communication. The following suggestions are made for appropriate modifications and publication:

It is best to provide a schematic diagram of reinforcement learning

How is the order of  zernike selected as 6?

Line 33, Figure???

The content in the article should be appropriately simplified, with common sense and unnecessary conceptual concepts that do not require excessive explanation

Author Response

Thanks for taking the time to review the manuscript. A point-by-point response follows. The reviewer's comments are italicized, while the responses are in regular font. In the revised manuscript, modified text in response to reviewer's comments is highlighted in blue font to facilitate its identification.

The paper analyzes the Reinforcement Learning environment of wavefront sensorless adaptive optics in the downlink of single mode fiber coupled satellite communication. The following suggestions are made for appropriate modifications and publication:

  1. It is best to provide a schematic diagram of reinforcement learning.
    Figure 1 in the revised manuscript has been modified to better show the schematic diagram of RL, including the relationship between the environment, agent, observation, reward, and action.
  2.  How is the order of  zernike selected as 6?
    In the revised manuscript, we modified the original text as follows to explicitly state the rationale behind the choice of the order of Zernike polynomials:
    "Our RL environment presents two choices within the action-space: the number of actuators or Zernike polynomials in deformable mirror. Given our available setup, the action-space can be configured with either directly 64-actuators or with deformable mirror Zernike polynomials truncated at the first 6-modes (2nd radial orders) in a circular geometry. The decision to use 6-mode Zernike polynomials stemmed from experiments conducted with Shack-Hartmann wavefront sensor-based AO. These experiments demonstrated that employing 6-modes of Zernike polynomials is adequate to achieve a Strehl ratio of more than 50% under average atmospheric turbulence conditions (where D/r_0 < 5)."
  3. Line 33, Figure???
    The text referred to a figure that was removed from the manuscript. Corrected in the revised version.
  4. The content in the article should be appropriately simplified, with common sense and unnecessary conceptual concepts that do not require excessive explanation.
    The revised manuscript has been trimmed of unnecessary parts referring to classical and well established material, for which a citation would suffice. This process had to be balanced against the the requests of other reviewers about adding details to some concept introduced in the paper. The content of section 2.3 on RL has beer removed, and condensed into the introduction, preserving the citations from the original version of the manuscript.

Reviewer 2 Report

Comments and Suggestions for Authors

The paper describes a number of interesting results obtained from a simulation work using a number of available software packages to examine the applicability of reinforced learning to the wavefront correction in free-space optical communications. The future of AI in the control of adaptive optics systems is enormous, so research proposals as the one present are extremely interesting.

However there are a number of issues that must be improved, to my understanding, in order to make the situation much closer to the one existing in practical applications. The quasi-static approach used by the authors is too far form reality and drives to conclusions without any practical use.

I would encourage authors to dig deeper in the problem and even to present in the future results coming from a laboratory test, at least.

Authors may find here a number of comments associated with the relevant line of the paper.  

33. The figure number is missing. Two question marks instead (broken link)

34. The figure 1 should be improved. Red lines showing the shape of the wavefront will NOT be flat after the imaging lens depicted (blue). They will be focusing instead. Traditional control systems do have a wavefront sensor accessed through a beamsplitter or dichroic, which is missing. Some controls may used the image at the focal plane, but they are not “traditional” as stated In the box. The converging lines drawn towards the Single-Mode Fibre are really the role of the imaging lens.

120. Typo in the word “deformable”

143. the figure of 1 ms is reasonable, but unfortunately it is highly dependent on the atmospheric conditions existing at the Optical Ground Station. The situation will be very different if the OGS is located at sea level or in an astrophysics observatory. The figure of 1ms is presented as an absolute value, which is not to my understanding.

145. The sentence “adjustment of the deformable mirror must be made within tens of seconds” is wrong and opposite to the concept of previous line. The deformable mirror must be run at a few thousand hertz in order to achieve the 1ms correction.

167. Figure 2 lacks the specification of the units of the axis, and also the one of the colour scale. Probably a number of considerations are required to obtain them from the simulation software, which may also be included in the text.

167. I don’t understand how the reliability of the optical link can be separated from the image sharpness. Indeed there is no image as output of the system, just the power coupled to the fibre. Maybe some rephrase will help.

205. To add an order of magnitude of those time steps is essential to evaluate up to what point it is a practical approach.

221. The actuation range of the deformable mirror, 1 micron, is much smaller than the one provided by ALPAO for its mirrors, and also other manufacturers. It is also too small for the expected turbulence in optical comms. An explanation for this figure, or change, will be good.

226. Real deformable mirrors are never squared in actuators, because the beams and the telescopes are circular-shaped, and the actuators located at the corners are removed. I see that the simulations have been carried out with squared pupils and images, which can be used to gen some insight, but they should be moved to more realistic shapes as soon as possible.

250. Maybe it a typo, but the r0=0 will imply infinity turbulence, which is nonsense. The authors should explain better what they are expecting for the 5 cm to 15 cm.

297. Computation of the Strehl ratio is normally very fast. Probably the fastest. I don’t understand why the authors thinks differently.

305. The definition of the r2 reward seem a little bit “ad hoc”, as opposed to the Strehl ratio which extremely well known and widely used. Perhaps the use of the encircle energy, which is also a well-known quality parameter, may be tried.

 366. There seem to be values of Strehl ratio below 0 in figure 5. They can not exist by definition, so there must mistakes in the calculations or in the drawing.

412. The Shack hartmann is said to have 12 lenslets. Probably you mean 12x12 =144 lenslets.

471. It would be nice to state how many Zernikes are you employing, there are a few ways in which the Zernikes polynomials can be ordered.

487. It is important to state also how many lenslets are you using for the Shack-Hartmann sensor, especially considering that this number is normally tuned with de expected D/r0

529. There seem to be an important word missing after “corresponding”.

531. I don’t think there is any “unrealiability” in the wavefront sensing carried out by a Shack-Hartmann sensor, There are, of course, many sources of errors and limitations.

Author Response

Thanks for taking the time to review the manuscript. A point-by-point response follows. The reviewer's comments are italicized, while the responses are in regular font. In the revised manuscript, modified text in response to reviewer's comments is highlighted in blue font to facilitate its identification.

The paper describes a number of interesting results obtained from a simulation work using a number of available software packages to examine the applicability of reinforced learning to the wavefront correction in free-space optical communications. The future of AI in the control of adaptive optics systems is enormous, so research proposals as the one present are extremely interesting.

  • However there are a number of issues that must be improved, to my understanding, in order to make the situation much closer to the one existing in practical applications. The quasi-static approach used by the authors is too far form reality and drives to conclusions without any practical use.
    We agree that there are many situations where the quasi-static approach is not applicable due to the limited speed of the deformable mirror and the low coherence time of some challenging atmospheric conditions. However, there are applications where the quasi-static conditions are applicable, such as with photonic adaptive optics concepts which can operate at tens of kHz (Momen et al. SPIE), or locations and environmental conditions where there is a sufficiently high coherence time to support hundreds of RL loop iterations with a fast deformable mirror. We have added the following text in the Introduction section of the manuscript, to clarify that the quasi-static hypothesis is not just preliminary work towards dynamic environments, but it has practical applications:
    In the quasi-static environment, it is assumed that the turbulence profile is static during training, corresponding to the condition that the RL model convergence is significantly faster than the atmospheric coherence time.  Such a condition can be relevant in more novel photonic adaptive optics concepts [24] and in long ground layer turbulence coherence time conditions.

  • I would encourage authors to dig deeper in the problem and even to present in the future results coming from a laboratory test, at least. 
    Thanks for the remark. In the revised version of the manuscript, we added text in the Conclusion section, describing the link between this work and ongoing and future research directions, putting the work in the context of a multi-phase project that will involve also hardware implementation and operation in dynamic environments.
  • Authors may find here a number of comments associated with the relevant line of the paper.
    33 The figure number is missing. Two question marks instead (broken link).
    The text in the original manuscript referred to a figure that has been removed. Corrected in the revised version.

  • 34. The figure 1 should be improved. Red lines showing the shape of the wavefront will NOT be flat after the imaging lens depicted (blue). They will be focusing instead. Traditional control systems do have a wavefront sensor accessed through a beamsplitter or dichroic, which is missing. Some controls may used the image at the focal plane, but they are not “traditional” as stated In the box. The converging lines drawn towards the Single-Mode Fibre are really the role of the imaging lens.
    Thanks for the suggestions. Figure 1 has been modified to correct the mistakes in the schematic of the RL environment of the wavefront sensorless AO system. Also, some modifications have been made to better show the schematic diagram of RL, including the relationship between the environment, agent, observation, reward, and action.
  • Typo in the word “deformable”.
    Corrected.
  • 143. the figure of 1 ms is reasonable, but unfortunately it is highly dependent on the atmospheric conditions existing at the Optical Ground Station. The situation will be very different if the OGS is located at sea level or in an astrophysics observatory. The figure of 1ms is presented as an absolute value, which is not to my understanding.
    The reviewer is correct in pointing that the turbulence timescale is highly dependent on the atmospheric conditions and is not an absolute value.   The 1 ms was chosen as a approximate number to represent the timescales at near sea-level conditions at high satellite elevation angles. In contrast to astronomical observatories at such high altitudes, the low altitude at ground stations leads to beams that experience turbulence near the ground level, which is slower with a coherence time of ~5ms, and turbulence at high altitudes (jet stream), which is faster at coherence time of ~0.2 ms. The ground-level turbulence is expected to be the dominant contribution for optical satcom and therefore we use 1 ms as a reasonable time-scale beyond most of the turbulence correction can be applied. We acknowledge that we are neglecting the correction of higher frequency components in our model by limiting us to 1 ms. 

    We clarified this position in the manuscript, by modifying as follows the text in Section 2.2:
    The atmosphere has a characteristic turbulence timescale on the order of ~1 ms, which varies significantly with the satellite’s elevation angle and turbulence conditions. This value can be as short as 0.2 ms, and as high as 10 ms. At the lowest elevations, the turbulence is the strongest, due to the effective thickness of the atmosphere. The ground layer turbulence has the longest coherence time, and the high altitude jet stream timescales are very fast. The wavefront is also rapidly changing based on the change of optical path resulting in the turbulence profile appearing to translate across the aperture as the telescope tracks the satellite, which is the dominant effect of the wavefront change at high satellite elevation angles. This means that the model must be able to correct the wavefront distortion within 1 ms to maintain a high degree of correction as the wavefront changes while the satellite passes. If an approximate solution can be found within a few milliseconds, the atmosphere can be considered to be in a quasi-static state and a static turbulence profile for the purposes of training can be considered valid. This paper is focused on optical ground stations near sea-level, and therefore we adopt 1 ms as the target coherence timescale, neglecting the weaker contribution of the high altitude winds. This applies to ground layer turbulence that evolves slower than 1 ms.

  • 145. The sentence “adjustment of the deformable mirror must be made within tens of seconds” is wrong and opposite to the concept of previous line. The deformable mirror must be run at a few thousand hertz in order to achieve the 1ms correction.
    Thanks. Typo corrected in the revised version.
  • 167. Figure 2 lacks the specification of the units of the axis, and also the one of the colour scale. Probably a number of considerations are required to obtain them from the simulation software, which may also be included in the text.
    Axes specifications and colour bars added in the revised manuscript.
  • I don’t understand how the reliability of the optical link can be separated from the image sharpness. Indeed there is no image as output of the system, just the power coupled to the fibre. Maybe some rephrase will help.
    The following text has been added in Section 2.3 to better explain the difference between image sharpness and optical link:
    "However, the existing implementations of RL on AO systems are not optimized for optical satellite communication due to their primary focus on optimizing image sharpness rather than improving the reliability of optical data links. In other words, although image sharpness can be used for fiber coupling efficiency, it gathers excessive information that can compromise loop speed. Ensuring a stable and dependable optical link is prioritized in towards a consistent and uninterrupted high-data transmission with lower latency and cost. A low pixel count detector mitigates high-latency and expensive read-out circuits, such as those found in infrared cameras. Additionally, image-sharpness based implementations are specifically customized for optimizing wavefront distortion under distinct conditions found in environments like microscopy and ophthalmology."
  • 205. To add an order of magnitude of those time steps is essential to evaluate up to what point it is a practical approach.
    We modified the sentence related to episode lengths as follows:
    “... To achieve this, we analyzed the various lengths of episodes, ranging from 10 to 100 time steps. Our findings indicate that an episode length of 20-30 time steps is sufficient to achieve optimal action.” 
  • 221. The actuation range of the deformable mirror, 1 micron, is much smaller than the one provided by ALPAO for its mirrors, and also other manufacturers. It is also too small for the expected turbulence in optical comms. An explanation for this figure, or change, will be good.
    Thanks for spotting this error. We checked the ALPAO DM data sheet and fixed this error. We indeed do not limit the actuation/stoke range to 1um. For optical satellite communication ground station telescopes of size ~0.5 m, there is a fast-steering mirror for precise pointing correction that will also correct for the large tip-tilt wavefront error contributions, significantly reducing the need for large stroke on the deformable mirror. The combination of the small aperture, the longer wavelengths, and the fast steering mirror compensating for most of the wavefront error by itself, leads to D/r0 values of less than 10, and the stroke will be limited to around +/- 5 um.
  • 226. Real deformable mirrors are never squared in actuators, because the beams and the telescopes are circular-shaped, and the actuators located at the corners are removed. I see that the simulations have been carried out with squared pupils and images, which can be used to gen some insight, but they should be moved to more realistic shapes as soon as possible.
    Indeed, the deformable mirrors are circular in shape and the corners are removed. We use circular aperture in all simulations, including the deformable mirror and when using a simulated Shack-Hartmann wavefront sensor to benchmark our performance. We have clarified in Figure 2 that the apertures have this circular geometry.
  • 250. Maybe it a typo, but the r0=0 will imply infinity turbulence, which is nonsense. The authors should explain better what they are expecting for the 5 cm to 15 cm.
    Corrected in the revised manuscript. Thanks.
  • 297. Computation of the Strehl ratio is normally very fast. Probably the fastest. I don’t understand why the authors thinks differently.
    To clarify this point, the following text has been added to the revised manuscript:
    “Although the Strehl ratio is commonly used in AO, its computation is not ideal for the applications in this work, since it requires a point spread function and a camera, whereas our system explicitly avoids reliance on a camera (specifically an InGaAs camera), in favour of a cheaper, compact, and faster solution. The calculation of the Strehl ratio requires sufficiently long exposures for speckle variation, and there are alternative methods that utilize fewer array sizes. Additionally, the Strehl ratio is not an ideal metric for coupling into a single-mode fiber, as the correlation between the single-mode fiber coupling efficiency and the Strehl ratio is reduced at high Strehl ratios, restricting its applicability as reward function at lower values. At higher Strehl ratios, it is limited by the mode mismatch between a focused flat top beam and the Gaussian mode in a single-mode fiber [56.57]. This suggests that the performance of a wavefront sensorless approach to adaptive optics is sensitive to feedback speed and latency, for which the Strehl ratio is not optimal. 
  • 305. The definition of the r2 reward seem a little bit “ad hoc”, as opposed to the Strehl ratio which extremely well known and widely used. Perhaps the use of the encircle energy, which is also a well-known quality parameter, may be tried.
    Please refer to the response to line 297 regarding our choice not to use the Strehl ratio. The reward r2 was formulated as a reward function encoding the alignment, beam width, and single mode fiber coupling efficiency information without requiring a full focal plane image. The use of encircled energy would provide insufficient information in the exploration stage. To further clarify, the following text was added to Section 3.4:
    "Although the total power of a single-mode fiber is a crucial indicator of its ability to concentrate power, due to its very small diameter, it is not particularly useful for exploration as it misses a significant portion of observations. Therefore, relying solely on the total power of a single-mode fiber as a reward function is inadequate. To address this, we combine it with SSIM as a criterion to measure the similarity between two arrays on a photodetector, like Strehl ratio. The first array serves as a reference, which represents the power distribution when there is no wavefront distortion, and the second array represents the current power distribution. In other words, single-mode fiber total power measures the performance of the exit of the single-mode fiber, while SSIM measures the performance of the entrance of the single-mode fiber."
  • 366. There seem to be values of Strehl ratio below 0 in figure 5. They can not exist by definition, so there must mistakes in the calculations or in the drawing.
    The shaded area in the figure reaches physically incompatible negative values in some regions of the figures mentioned by the reviewer because the standard deviation may be larger than the value of the mean with respect to 0. In order to clarify this point, we added the following note to the captions of the figures affected by this:
    "Note that the shaded regions extend to negative values of the Strehl ratio because the standard deviation may be larger than the mean, represented by the solid curves."
  • 412. The Shack hartmann is said to have 12 lenslets. Probably you mean 12x12 =144 lenslets.
    Thanks for spotting this error. 12 is the number of lenslets across the diameter of the aperture, which is circular. We clarified this in the manuscript by adding the following text in Section 5.1:
    "Employing the Shack-Hartmann wavefront sensor with 12 lenslets across the aperture diameter for a total of 112 lenslets as a benchmark enabled us to comprehensively...” 
  • 471. It would be nice to state how many Zernikes are you employing, there are a few ways in which the Zernikes polynomials can be ordered.
    Through the revised manuscript, the text "6th order Zernike series" has been changed to "first 6-modes (2nd radial order) Zernike polynomials".
  • It is important to state also how many lenslets are you using for the Shack-Hartmann sensor, especially considering that this number is normally tuned with de expected D/r0.
    We added the number of lenslets was added, as specified above.
  • There seem to be an important word missing after “corresponding”.
    Thanks for spotting the error. It is fixed in the revised manuscript.
  • 531. I don’t think there is any “unrealiability” in the wavefront sensing carried out by a Shack-Hartmann sensor, There are, of course, many sources of errors and limitations.
    We agree that the phrasing in the original manuscript was not clear and lacked precision. We modified the text as 
    “For existing and planned optical ground stations, AO may not be considered at all due to the cost, complexity, calibration, wavefront steps, aliasing, and latencies involved in Shack-Hartmann wavefront sensing.” 

Reviewer 3 Report

Comments and Suggestions for Authors

This study explores using reinforcement learning (RL) as a cost-effective alternative to traditional wavefront sensor-based solutions for Optical Satellite Communications (OSC) downlinks. The topic is interesting, and the paper makes a significant contribution to the field, here are some suggestions to the authors:

·        line 33 the numbering of the figure is not shown correctly.

·        Authors are recommended to add updated references from 2022 and 2023 that enrich the introduction.

·        In the conclusion suggesting future research directions that focus on real-time algorithm optimization and integration into live OSC systems could provide a more comprehensive outlook on the practical implications of your findings.

·        It would be interesting if the authors discussed future work where the research could be directed.

Comments on the Quality of English Language

Minor editing of English language required

Author Response

Thanks for taking the time to review the manuscript. A point-by-point response follows. The reviewer's comments are italicized, while the responses are in regular font. In the revised manuscript, modified text in response to reviewer's comments is highlighted in blue font to facilitate its identification.

This study explores using reinforcement learning (RL) as a cost-effective alternative to traditional wavefront sensor-based solutions for Optical Satellite Communications (OSC) downlinks. The topic is interesting, and the paper makes a significant contribution to the field, here are some suggestions to the authors:

  • line 33 the numbering of the figure is not shown correctly.
    The cross reference in the original manuscript referred to a figure that was removed. Corrected in the revised version.
  • Authors are recommended to add updated references from 2022 and 2023 that enrich the introduction.
    We added 10 references from 2022-2023 and removed 2 less recent ones.
  • In the conclusion suggesting future research directions that focus on real-time algorithm optimization and integration into live OSC systems could provide a more comprehensive outlook on the practical implications of your findings.
    The part that explains the future research directions has been improved and presented more comprehensively. Please check the Future Work part of the Conclusion, where the following text has been added:
    "Future work aims at improving satellite communication downlinks, seeking a faster, more stable, reliable, and cost-effective optical link. Cost savings of 30% to 40% are expected through the use of smaller telescope systems and without the need for Shack-Hartmann wavefront sensors. This can be achieved by developing the framework of a deep RL model and training it to adapt to dynamically changing wavefront distortions across different satellite trajectories and deformable mirror actuator geometries. Our immediate focus involves investigating the performance of Deep RL algorithms under various dynamic turbulence conditions within a simulated environment. Once this challenge is addressed, our next step involves training our model using the available setup, omitting wavefront sensors and cameras but using low-pixel photodetectors instead."
  • Minor editing of English language required.
    We thoroughly proof read the revised manuscript, correcting all the typos we could find.

Reviewer 4 Report

Comments and Suggestions for Authors

Comment 1

While the abstract provides a clear and informative overview of the research, there are some potential negative point that could be improved:

-The abstract could benefit from more specific details about the RL-based approach and the 2x2 photodiode array used. Providing a bit more technical information could help the reader understand the research's novelty and innovation.

Comment 2

-The abstract lacks a clear concluding statement that summarizes the key takeaways of the research and its implications. A strong conclusion can leave a lasting impression on the reader.

Comment 3

-Using technical terms like "Proximal Policy Optimization (PPO)" and "Soft Actor-Critic (SAC)" without providing explanations. Depending on the target audience, it might be helpful to provide brief explanations or references to these terms.

Comment 4

-The introduction to the paper provides a clear outline of the research program's three phases and highlights the focus on reinforcement learning (RL) for wavefront sensorless adaptive optics (AO). While the introduction is informative and well-structured, there are a few potential negative points:

-While you mention the contributions of the work, they are somewhat broad and lack specific details. For example, it could be more precise about what makes the simulated wavefront sensorless AO-RL environment unique and innovative.

- The introduction refers to several technical terms and acronyms, such as "Shack-Hartmann wavefront sensor" and the RL algorithms, without providing explanations. This might assume prior knowledge from the readers.

Comment 5

-In section 3.1, while you mention episode durations (e.g., 20, 30, and 50 time steps), you do not explain how these durations were chosen or if they have specific significance in the research. Providing some quantitative reasoning can add depth to the explanation.

Comment 6

-In section 4.3 the explanation of the "clipped surrogate objective" and the policy regulation technique relies on references "[20]". It would be beneficial to provide a bit more context or brief explanations of these concepts to make it clear in your manuscript.

Comment 7

-In the conclusion section you mention that the PPO algorithm is effective in achieving a high average Strehl ratio, but you do not provide specific numerical results or performance metrics. Including quantitative data would make the conclusion more convincing.

-In the same section you mention challenges related to partial observability and the curse of dimensionality when transitioning to a semi-dynamic environment. However, you do not delve into these challenges in detail or offer insights into how they might be addressed.

-The mention of "investigating the performance of RL algorithms for various dynamic turbulence conditions at different times of the day while the satellite is not fixed on the sky" is somewhat vague. Providing more specific details about the goals and objectives of future work would be beneficial.

- The conclusion could benefit from a discussion of the practical applications of the research and its potential impact on optical satellite communication. How might the findings be applied in real-world scenarios?

Comments on the Quality of English Language

Overall, the quality of English language is good, but ensuring that technical terms and concepts are well-explained, avoiding ambiguity, and providing supporting data or examples can further enhance the clarity and readability of the paper.

Author Response

Thanks for taking the time to review the manuscript. A point-by-point response follows. The reviewer's comments are italicized, while the responses are in regular font. In the revised manuscript, modified text in response to reviewer's comments is highlighted in blue font to facilitate its identification.

  1. While the abstract provides a clear and informative overview of the research, there are some potential negative point that could be improved:

    - The abstract could benefit from more specific details about the RL-based approach and the 2x2 photodiode array used. Providing a bit more technical information could help the reader understand the research's novelty and innovation.
    The following sentence was added to give more details about the RL-based approach:
    To gain a deeper insight into these challenges, we have developed and shared the first OSC downlinks RL environment
    and evaluated a diverse set of deep RL algorithms in the environment.

  2. The abstract lacks a clear concluding statement that summarizes the key takeaways of the research and its implications. A strong conclusion can leave a lasting impression on the reader.
    The following sentence was added at the end of the abstract to improve the concluding statement:
    Our findings indicate the potential of RL in replacing wavefront-based AO while reducing the cost of OSC downlinks. 
  3. Using technical terms like "Proximal Policy Optimization (PPO)" and "Soft Actor-Critic (SAC)" without providing explanations. Depending on the target audience, it might be helpful to provide brief explanations or references to these terms.
    To provide a brief explanation of RL algorithms and Shack-Hartmann, we modified part of the abstract as follows
    To gain a deeper insight into these challenges, we have developed and shared the first OSC downlinks RL environment and evaluated a diverse set of deep RL algorithms in the environment. Our results indicate that the Proximal Policy Optimization (PPO) algorithm outperforms the Soft Actor-Critic (SAC) and Deep Deterministic Policy Gradient (DDPG) algorithms. Moreover, PPO converges to within 86% of the maximum performance achievable by the predominant Shack-Hartmann wavefront sensor-based AO system. Our findings indicate the potential of RL in replacing wavefront sensor-based AO while reducing the cost of OSC downlinks.
  4. The introduction to the paper provides a clear outline of the research program's three phases and highlights the focus on reinforcement learning (RL) for wavefront sensorless adaptive optics (AO). While the introduction is informative and well-structured, there are a few potential negative points:
    1. While you mention the contributions of the work, they are somewhat broad and lack specific details. For example, it could be more precise about what makes the simulated wavefront sensorless AO-RL environment unique and innovative.
      We added the following text in the Introduction to give more detail about our RL environment
      .... .
      This is the first AO-RL environment that is implemented according to the standards of the OpenAI Gymnasium framework which simplifies the analysis of RL algorithms. The related source code link can be found in the `Data Availability Statement' section.” 
    2. The introduction refers to several technical terms and acronyms, such as "Shack-Hartmann wavefront sensor" and the RL algorithms, without providing explanations. This might assume prior knowledge from the readers.
      A description of the Shack-Hartmann wavefront sensor has been added in the introduction:
      The Shack-Hartmann wavefront sensor is a commonly used sensor that operates by using an aperture equipped with small lenses, referred to as the lenslet arrays. This sensor measures the displacement of the focal spots as a distorted wavefront propagates through the lenslet array [25]”.
      Detailed explanations of RL algorithms are given in the dedicated sections; however, to improve the readability of the Introduction, we added the following text:
      “In our empirical analysis, we compare SAC, DDPG and PPO deep RL algorithms to an idealized traditional AO system with a Shack-Hartmann wavefront sensor. For further details on the Deep RL algorithms mentioned, please refer to sections 4.1, 4.2 and 4.3, respectively.”  
  5. In section 3.1, while you mention episode durations (e.g., 20, 30, and 50 time steps), you do not explain how these durations were chosen or if they have specific significance in the research. Providing some quantitative reasoning can add depth to the explanation.
    To clarify this point, we modified the sentence related to episode length as follows:
    “We conducted a parametric study with respect to the the episode length, ranging from 10 to 100 time steps. Our findings indicate that the adoption of an episode length of 20-30 time steps is sufficient to achieve optimal action.

  6. In section 4.3 the explanation of the "clipped surrogate objective" and the policy regulation technique relies on references "[20]". It would be beneficial to provide a bit more context or brief explanations of these concepts to make it clear in your manuscript.
    To further clarify the properties of PPO, we expanded the text in section 4.3,  also adding equation (4) that shows the expression of the objective function.

  7. Conclusion
    1. In the conclusion section you mention that the PPO algorithm is effective in achieving a high average Strehl ratio, but you do not provide specific numerical results or performance metrics. Including quantitative data would make the conclusion more convincing.
      The section "Conclusion" has been expanded, adding numerical results of PPO in quasi-static and semi-dynamic environments, with detailed discussion and comparison.
    2.  In the same section you mention challenges related to partial observability and the curse of dimensionality when transitioning to a semi-dynamic environment. However, you do not delve into these challenges in detail or offer insights into how they might be addressed.
      In section 5.2, we have the following discussion that should clarify and address the point raised by the reviewer:
      "...the configuration used for a quasi-static environment is insufficient for learning a policy within a semi-dynamic environment. Based on this poor performance, we hypothesize that partial observability and the curse of dimensionality can be significant sources of this problem. The observation-space is limited to a low dimension of 2 x 2 pixels, which may result in inadequate information gathered from the environment, leading to partial observability. Additionally, the utilization of a high-dimensional action-space can cause an exponential rise of computational effort and lead to the curse of dimensionality."

    3. The mention of "investigating the performance of RL algorithms for various dynamic turbulence conditions at different times of the day while the satellite is not fixed on the sky" is somewhat vague. Providing more specific details about the goals and objectives of future work would be beneficial.
      We expanded the description of Future Work in the section Conclusion, including also clarifying details about this point:
      "Future work aims at improving satellite communication downlinks, seeking a faster, more stable, reliable, and cost-effective optical link. Cost savings of 30% to 40% are expected through the use of smaller telescope systems and without the need for Shack-Hartmann wavefront sensors. This can be achieved by developing the framework of a deep RL model and training it to adapt to dynamically changing wavefront distortions across different satellite trajectories and deformable mirror actuator geometries."
    4. The conclusion could benefit from a discussion of the practical applications of the research and its potential impact on optical satellite communication. How might the findings be applied in real-world scenarios?
      In addition to the text above, we added the following:
      "Our immediate focus involves investigating the performance of Deep RL algorithms under various dynamic turbulence conditions within a simulated environment. Once this challenge is addressed, our next step involves training our model using the available setup, omitting wavefront sensors and cameras but using low-pixel photodetectors instead."

Overall, the quality of English language is good, but ensuring that technical terms and concepts are well-explained, avoiding ambiguity, and providing supporting data or examples can further enhance the clarity and readability of the paper.
We thoroughly proof read the revised manuscript, policing the English and correcting as many typos as we could detect.

Round 2

Reviewer 4 Report

Comments and Suggestions for Authors

The authors have addressed most of comments satisfactorily.